# ONLY-IF: REVEALING THE DECISIVE EFFECT OF INSTRUCTION DIVERSITY ON GENERALIZATION

## ABSTRACT

Understanding and accurately following instructions is critical for large language models (LLMs) to be effective across diverse tasks. In this work, we conduct a rigorous investigation into the factors that enable generalization to unseen instructions. Through controlled experiments, inspired by the Turing-complete Markov algorithm, we demonstrate that such generalization **only emerges** when training data is diversified enough across semantic domains. Our findings also reveal that merely diversifying within limited domains fails to ensure robust generalization. In contrast, cross-domain data diversification, even under constrained data budgets, significantly enhances a model's adaptability. We further extend our analysis to real-world scenarios, including fine-tuning of *specialist* and *generalist* models. Our research provides important insights for dataset collation, particularly when optimizing model performance by expanding training data for both specialist and generalist scenarios. We show that careful consideration of data diversification is key: training specialist models with data extending beyond their core domain leads to significant performance improvements, while generalist models benefit from diverse data mixtures that enhance their overall instruction-following capabilities across a wide range of applications. Our results highlight the critical role of strategic diversification and offer clear guidelines for improving data quality.

## 1 INTRODUCTION

The rapid advancements in large language models (LLMs) have revolutionized a wide range of tasks, including language comprehension (Wang et al., 2020), generation (Brown et al., 2020), knowledge-based question answering (Hendrycks et al., 2021a), and solving complex reasoning problems in fields like mathematics (Cobbe et al., 2021; Hendrycks et al., 2021b) and programming (Chen et al., 2021a; Austin et al., 2021; Chen et al., 2021b; Li et al., 2022). These successes hinge on the foundational capabilities of LLMs, such as knowledge retrieval, reasoning, planning, and notably, instruction following—enabled through instruction-tuning (Ouyang et al., 2022; Taori et al., 2023; Wei et al., 2022; Sanh et al., 2022a). Instruction-tuning trains LLMs on a wide variety of instruction-output pairs, allowing them to handle diverse prompts and better generalize to new tasks.

While knowledge retrieval focuses on accessing stored information and reasoning involves multi-step problem-solving, instruction following concerns the accurate interpretation and execution of diverse natural language prompts (Zhou et al., 2023b). This capability is vital for user interaction, as it involves understanding the intent behind instructions and performing tasks without relying on complex logic (Liu et al., 2024a). Despite its importance, the mechanisms underlying instruction following remain less explored compared to other capabilities like reasoning.

The current research landscape on instruction tuning has produced varied and sometimes contradictory findings, ranging from the impact of dataset selection (Zhou et al., 2023a) to the effects of scaling up data (Zeng et al., 2024; Zhang et al., 2024a). These disparate findings suggest that instruction following in LLMs is influenced by the scale and composition of fine-tuning data in complex ways Dong et al. (2024); Zhang et al. (2024b). However, a systematic investigation into the core components of LLM performance—specifically isolating instruction following from reasoning and knowledge retrieval—has been limited.

Our work addresses this gap by focusing explicitly **ONLY** on the **I**nstruction-**F**ollowing capabilities of LLMs. We first introduce a systematic analysis of instruction diversity through a controlled

symbolic task—string rewrites—inspired by the Turing-complete Markov algorithm Markov (1954). By isolating the effects of instruction diversity, we focus on the model's ability to follow instructions without conflating this with reasoning capabilities. Through controlled experiments, we examine the impact of instruction diversification across various semantic domains on the model's ability to adapt to unseen instruction semantics. We find that generalization to unseen task semantics emerges **ONLY IF** the instructions are sufficiently diversified. Our findings reveal that diversification confined to limited domains does not guarantee robust generalization. In contrast, cross-domain diversification significantly enhances the model's adaptability to new instructions, highlighting the importance of a more diverse training strategy.

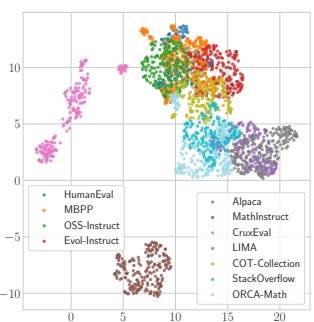 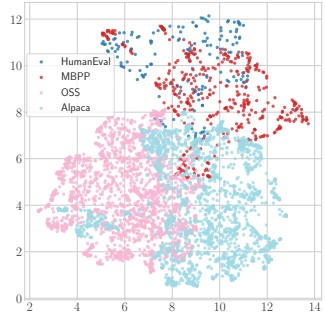 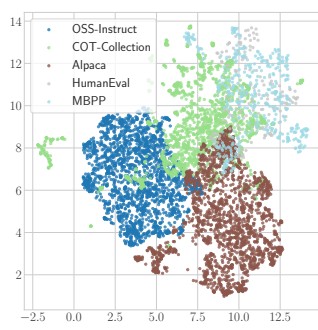

(a) Semantic clustering of relevant datasets.

(b) OSS-Alpaca mixture and test instructions.

(c) OSS-COT-Alpaca mixture and test instructions.

Figure 1: Visualization of embedded instructions.

Additionally, our research offers and empirically verifies key insights into dataset collation strategies for both **specialist** and **generalist** LLMs. We show that when training **specialist** models (e.g., code language models), extending data diversification beyond their core domain yields substantial performance improvements in instruction following. For **generalist** models, using diverse data mixtures enhances their instruction-following capabilities across a broad range of applications, even for domains they were not explicitly trained on. These findings emphasize the importance of diversification in optimizing LLM performance and adaptability. Our study provides insights for dataset collation, especially when ***dataset expansion*** is needed, showing how proper data diversification boosts performance in both specialist and generalist LLM scenarios, and is more effective than even multiplying dataset sizes.

## 2 EXPERIMENTS WITH STRING REPLACEMENTS

### 2.1 STRING REPLACEMENT TASK

To start a systematic investigation of instruction-following, we model instruction-following tasks as string-replacement operations, which are fundamental in theoretical computer science. String replacement forms the basis of Markov Algorithms (Markov, 1954), a Turing-complete model where sequences are iteratively transformed using ordered rewrite rules. These algorithms represent a structured, rule-driven process in which the first applicable rule is used to replace the leftmost matching substring, continuing until no further matches exist. This structured transformation mirrors the sequential, instruction-driven behavior we aim to explore. Appendix A provides illustrative examples of Markov algorithms in action.

In our study, we focus on a simplified form of this process. The replacement rule $R$—a pair like $aa \rightarrow bac$—is applied to an input string $\xi$ (e.g., $caaba$), yielding an output string $\tau$ (e.g., $cbacba$). The rule is applied to the leftmost occurrence of a match, and if no match exists, the original input remains unchanged. For instance, applying $\iota : iss \rightarrow art$ to $\xi = $ `mississippi` produces $\tau = $ `martissippi`, while applying $\iota$ to $\xi = $ `canada` leaves the string unaltered.

This task, though simple, serves as a powerful proxy for instruction tuning by teaching models to handle structured, rule-based transformations. Training on these string rewrites allows models

to internalize core aspects of instruction-following tasks, equipping them with the capability to generalize across a wide range of rule-based tasks without the overhead of more complex operations.

## 2.2 EXPERIMENT SET-UP

We designed two string replacement tasks to evaluate the model's ability to follow simple transformation rules:

**Basic Replacement**: Apply the rule $\iota : x \to y$ to any input string containing $x$. The model replaces the first occurrence of $x$ with $y$ and returns the modified string; and

**Conditional Replacement**: Apply the rule $\iota : x \to y$ if $x$ is present in the input string; otherwise, return the input unchanged.

The model takes in $\xi$ and $\iota$, outputs either the transformed string or the original input if no match is found.

Task 1 focuses on fundamental rewrite operations, while Task 2 introduces a conditional decision-making aspect. This experimental setup allows us to assess the model's capacity for both direct rule application and handling cases where no action is required.

In later sections, we also generalize the task from regular to context-sensitive, where $x$ and $y$ represent abstract structures e.g. $x = a^2$ will represent all squared terms.

We train GPT-2 models (256 dimensions, 6 layers, 4 attention heads) on synthetic instruction/outcome pairs. The dataset contains $S \times I$ sequences, with $S$ input strings and $I$ replacement rules. Models are tested on $10^5$ unseen examples to assess generalization across rules. Full training details are in Appendix B.

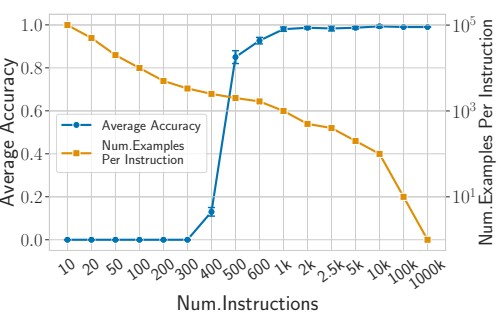

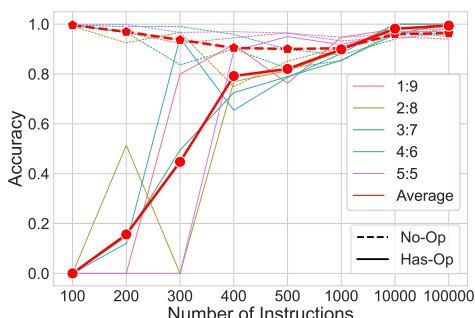

(a) Re-writing accuracy against the number of instructions with a fixed-size training set.

(b) Rewriting with no-op situation included.

Figure 2: Generalization versus the number of instructions during training.

## 2.3 RESULTS

### 2.3.1 INSTRUCTION DIVERSITY IS DECISIVE TO GENERALIZATION

Figure 2a presents the generalization accuracy for models trained on a fixed budget of $I \times S = 10^6$ examples, as a function of the number $I$ of different instructions in the training set. The number of examples per instruction ($S$) decreases as $I$ increases. In these experiments, all input sequences feature at least one instance of the replacement rule.

In Figure 2a, a sharp step-shaped transition is observed: models trained on fewer than $I_{min}$ (where $I_{min} = 300$) unique instructions consistently fail to generalize, regardless of example count per instruction. Conversely, models exposed to over $I_{max}$ (where $I_{max} = 1,000$) distinct instructions generalize effectively to unseen instructions, even with minimal examples per instruction. This phase transition underscores our hypothesis that the sheer number of distinct instructions ($I$) is a key driver for generalization, affirming the necessity of cross-domain diversification for robust instruction-following for unseen instructions.

### 2.3.2 DIVERSIFICATION ALLOWS GENERALIZATION IN CASE-BASED REASONING SET-UP

In earlier experiments, the model replaced a sub-string always present in the input. Now, we introduce a more complex task where some rules may not apply, requiring the model to decide whether a rule is applicable and either perform the operation or leave the input unchanged. This two-step process challenges the model to determine rule applicability and execute the correct action simultaneously.

To explore this, we introduce a third parameter in the training set: the frequency of "No-Ops" (instructions that cannot be satisfied), which we vary between 10% and 50%. The size of the training and test sets remains the same as in previous experiments, keeping the data size constant.

Figure 2b presents the generalization accuracies of trained models, as a function of the number of instructions and the frequency of No-Ops. Interestingly, despite No-Ops dominating the dataset[1], the model generalizes well to unseen instructions after training on around 400 distinct cases. The proportion of No-Ops does not significantly affect generalization beyond that point, demonstrating that training on diverse instructions effectively teach the model to assess rule relevance and apply them accurately.

Overall, our conclusions remain consistent with previous experiments, albeit with a slightly lower number of instructions needed for generalization (400 vs. 500). This demonstrates the effectiveness of diversification in more complex scenarios involving case-based reasoning and rule relevance assessment.

### 2.3.3 IMBALANCED DISTRIBUTION IS STILL EFFECTIVE IN DRIVING GENERALIZATION

In previous experiments, instructions were evenly distributed between examples in the training set: in a training set of 1 million examples, with 500 different instructions, every instruction would be featured 2000 times. Such a situation is unlikely to happen in real-world settings. In real-world training sets, some tasks will be much more common than others (due to the availability of fine-tuning data and the nature of the tasks).

To investigate the impact of the distribution of instructions on generalization to unseen tasks, we generate datasets of 1,000, 10,000 and 100,000 different instructions, and distribute the number of examples per instruction according to a power law distribution with PMF $f(x) = \alpha x^{\alpha-1}$ where $\alpha$ is the shape parameter. By varying the shape parameter of the power law, we can generate a distribution of examples that range from close to uniform, to extremely peaked as shown in Fig. 7.

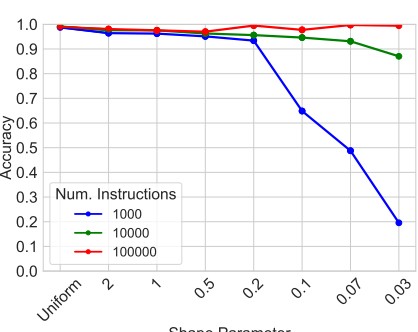

Figure 3 shows model generalization as a function of the power law's shape parameter for training sets of $N = 1$ million examples with $I_1 = 1,000$, $I_2 = 10,000$, and $I_3 = 100,000$ instructions. Models trained on $I_2$ or more instructions are robust to the distribution of examples per instruction. For models trained on $I_1$, generalization accuracy drops sharply when the shape parameter exceeds $\alpha = 0.2$. In such cases, instructions with a probability lower than $p_{low} = 0.1\%$ are barely represented, and the model effectively trains on fewer than $I_{min} = 500$ instructions, the minimum for generalizing to unseen instructions.

Figure 3: Effect of long-tail task distributions on model's generalization ability.

This observation suggests that generalization is achievable as long as there is sufficient semantic coverage, even if the data distribution is imbalanced. In practice, achieving uniform semantic coverage across the data may not be necessary to enable generalization. Instead, focusing on ensuring a broad enough range of semantics can still support effective model generalization, despite potential data imbalances.

---

[1]Consider a dataset containing 100,000 data points, 10% No-Ops, and 100 rules. No-Ops takes up 10,000 in total, ∼11× of any has-Ops.

### 2.3.4 SEMANTIC DIVERSIFICATION BOOSTS TASK PERFORMANCE

In real-world instruction-tuning, it is impractical to sample instructions uniformly from all semantic spaces due to their vastness. Instead, focusing on constrained sub-domains is crucial. We emulate this scenario by training models on instructions with semantic constraints, such as **repeated characters** ($aaabbbccc$ for $k = 3$ - each character repeated 3 times), **periodic patterns** ($abcabc$ for $k = 2$ a sub-string repeated 2 times), and **mirrored structures** ($abccbaabc$ for $k = 3$ mirroring the substring for 3 times), and measure their generalization across different levels of $k$, where $k$ is a parameter controlling the constrained-ness of rule semantics.

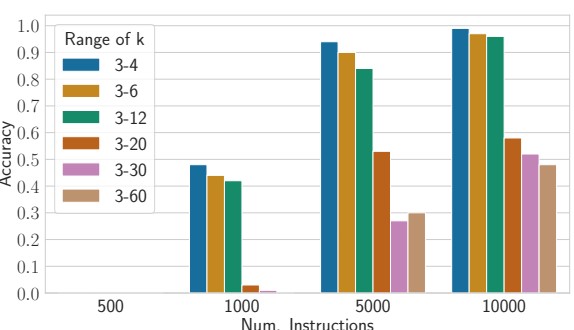

Our findings show that models trained on a single constrained sub-domain with high constraints (large $k$) fail to generalize to less constrained tasks (low $k$), indicating overfitting to specific patterns. Training on mixtures of constrained sets improves generalization, but only when the subspace is sufficiently rich. Larger instruction sets (e.g., 5000 examples) boost generalization, while highly restricted semantic domains (larger $k$) make it harder.

Figure 4: Model's performance on $k < 3$ when trained on the three classes of restricted semantics as in 2.3.4. Models trained on 500 or less instructions never generalize to smaller k.

These results underscore that instruction diversity must span a range of semantically rich sub-spaces, not just rely on larger datasets within highly restricted domains, to foster robust

generalization across tasks.

## 3 REWRITING WITH ABSTRACTION

In real-world applications, following instructions often extends beyond simple "search-and-replace" tasks, requiring a model to abstract high-level concepts and ground them in specific contexts.We therefore extend the string-rewriting experiment to simulate this challenge by including the aspect of abstraction and grounding into our set-up by distinguishing between abstract rules and their groundings to the input. This is crucial because abstract rules act as entry points into different semantic domains, each requiring unique reasoning strategies. This mirrors real-world tasks, such as translation and question answering, where the nature of the instructions varies widely depending on the task's context and domain-specific requirements.

Concretely, here we introduce a mathematical deduction task. Specifically, the model is tasked with simplifying algebraic expressions by applying specified deduction rules. Here, handling different abstract rules in mathematics is analogous to following instructions across distinct semantic domains.

In this experiment, we present the model with a randomly generated mathematical expression $I$ and an **abstract** mathematical deduction rule $\iota_{abs} : (X = Y)$ where $X$ and $Y$ are math expressions (e.g. an example abstract rule would be $a^2 - b^2 = (a + b) \times (a - b)$, to assess the model's ability to identify the relevant sub-expression in $\xi$ (e.g. in $(2x + 5)^2 - (3y - 6)^2$, here $a = 2x + 5, b = y - 6$ ) and correctly apply the transformation (get $(2x + 5 + 3y - 6) \times (2x + 5 - (y - 6))$). We observe how well the model generalizes when trained on datasets of varying rule diversity.

### 3.1 DATA GENERATION

We collate a set of distinct **equational** algebraic deduction rules of the form `LHS = RHS`.

**Random Tree Generation**   We randomly construct mathematical expression trees of a specified depth $d$. Non-leaf nodes were systematically assigned operators (e.g., $+, -, *, /$), while leaf nodes were populated with variables, constants, or unary operations.

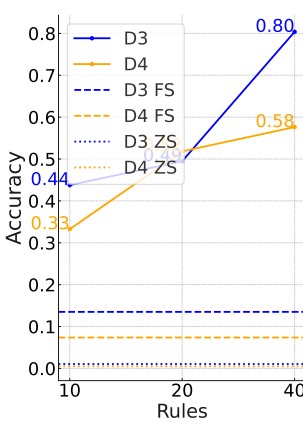 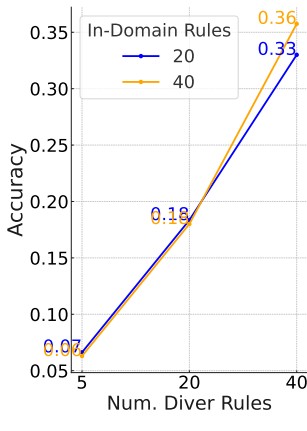 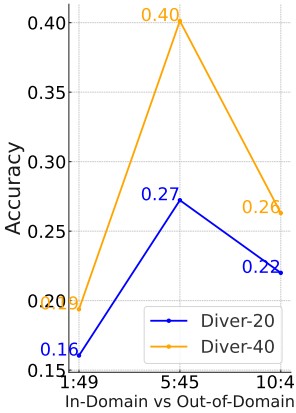

(a) Accuracy On Unseen Deduction Rules vs. Abstract Rule Diversity. **D3**:pattern with tree depth 3. **ZS**:Zero shot. **FS**: Few shot.

(b) Accuracy on unseen depths against number of diversification rules with the same in-domain / out-of-domain mixture.

(c) Accuracy on unseen depths vs. In-domain/out-of-domain combination. **Diver-** means number of diversification rules. X axis ticks are $|\mathcal{R}_{diver}|$: $|\mathcal{R}_{spec}|$.

Figure 5: Rewriting With Abstraction.

**Producing Pattern-Carrying Sub-Expressions**   To generate expressions, a pattern-carrying subtree was generated with a depth of $d_p$, denoting the depth of expression tree for each entry in the rule. e.g. we may replace $a$ with $(y + 2x + 5)$ with $d_p = 2$. We then randomly choose a leaf node and swap it with the **concrete** sub-expression.

### 3.2 SIMULATING SPECIALIST AND GENERALIST TRAINING

We examine two common settings for instruction-tuning: *generalist*, and *specialist* training via the set of simulated experiments.

#### 3.2.1 DIVERSIFYING INSTRUCTION SEMANTICS EMPOWERS BETTER GENERALIST LMs

We control the number of training instances (50K), and vary the number of abstract deduction rules & the instantiated rules per abstract rule. We test the models on unseen sequences and unseen abstract rules. We train the model on triples of $(\xi, \iota_{abs}, \tau)$ pairs where $\tau$ is the result of applying $\iota_{abs}$ to $\xi$. We fine-tune all models from a pre-trained Mistral-7B-v0.3 (Jiang et al., 2023) checkpoint. The result is shown in Figure 5a.Our findings demonstrate a clear advantage of increased rule diversity, consistent with previous string rewriting experiments. Holding the number of training instances constant, we observed that expanding the diversity of rules ($\iota$) in the training data significantly enhances the model's ability to generalize to unseen abstract rules during testing. This improvement is achieved despite the model encountering fewer groundings per rule and expression. These results not only validate the hypothesis from our string rewriting experiments but also underscore that increased diversity in training data significantly enhances the model's ability to concretize abstract instructions, improving generalization even with limited exposure to specific instances. This underscores the importance of diversity in developing models that can unseen tasks.

#### 3.2.2 SWEET POINT BETWEEN SPECIALIZATION AND DIVERSIFICATION FOR OPTIMAL SPECIALIST PERFORMANCE

In this experiment, we simulate a scenario where a model is trained as a specialist and tested on out-of-distribution queries that still belong to the same overall instruction type. To achieve this, we divide the set of rules $\mathcal{R}$ into two categories: specialized rules $\mathcal{R}_{spec}$ and diversification rules $\mathcal{R}_{diver} = \mathcal{R} \setminus \mathcal{R}_{spec}$. The training data is constructed as a mixture of instances generated from these two sets of rules. Specifically, the instances based on the specialized rules use patterns with depth $d_p^{train} < d_p^{test}$ to simulate out-of-distribution test instances, while the instances from the diversification

| Base Budget (OSS) | +OSS | +Alpaca | +CoT | HE | HE+ | MBPP | MBPP+ | Avg (Base) | Avg (+) | Avg | Rel. Gain |
|---|---|---|---|---|---|---|---|---|---|---|---|
| 15 | 5 | 0 | 0 | 62.2 | 56.7 | 75.7 | 62.4 | 68.9 | 59.5 | 64.2 | - |
| 15 | 4 | 1 | 0 | 67.1 | 59.8 | 75.7 | 62.2 | 71.4 | 61 | 66.2 | 3.1 |
| 15 | 3 | 2 | 0 | **68.3** | **61.6** | 75.4 | 63.2 | **71.9** | **62.4** | **67.1** | **4.6** |
| 15 | 2 | 3 | 0 | 64.6 | 60.4 | 76.4 | 63.4 | 70.5 | 61.9 | 66.2 | 3.1 |
| 15 | 1 | 4 | 0 | 65.2 | 58.5 | **76.7** | **63.7** | 71 | 61.1 | 66 | 2.8 |
| 15 | 0 | 5 | 0 | 64.6 | 57.9 | 73.9 | 61.4 | 69.3 | 59.7 | 64.5 | 0.5 |
| 60 | 15 | 0 | 0 | 66.5 | 61.6 | 75.4 | 61.9 | 71 | 61.8 | 66.4 | - |
| 60 | 0 | 15 | 0 | 67.1 | 59.8 | 77 | 64.8 | 72.1 | 62.3 | 67.2 | 1.2 |
| 60 | 0 | 7.5 | 7.5 | 64.6 | 59.1 | 76.2 | 63.8 | 70.4 | 61.5 | 65.9 | -0.6 |
| 60 | 7.5 | 7.5 | 0 | 68.9 | 61.6 | 76.2 | 63.8 | 72.6 | 62.7 | 67.6 | 1.9 |
| 60 | 7.5 | 3.25 | 3.25 | 66.5 | **62.2** | 76.2 | 64.3 | 71.4 | 63.3 | 67.3 | 1.4 |
| 60 | 12.5 | 2.5 | 0 | 68.3 | 61 | 76.2 | 64.3 | 72.3 | 62.7 | 67.5 | 1.6 |
| 60 | 12.5 | 1.25 | 1.25 | 68.3 | 62.2 | **77.8** | **65.1** | **73.1** | **63.7** | **68.4** | **3** |
| 60 | 14 | 1 | 0 | **69.5** | 62.2 | 76.2 | 64.3 | 72.9 | 63.3 | 68.1 | 2.5 |
| 60 | 14 | 0.5 | 0.5 | 66.5 | 61 | 76.5 | 63.2 | 71.5 | 62.1 | 66.8 | 0.7 |

Table 1: Results on DeepSeek-Coder-6.7B and comparison With MagiCoder-DS-6.7B Wei et al. (2023). Plum-colored row surpasses the performance of full-data training. Best configurations corresponding to each setting are highlighted. We demonstrated that one could achieve higher performances by means of diversification.

rules add variety to the training. This setup allows us to examine the trade-off between specialization and diversification for better instruction following in this specialized task.

Figure 5c, the results exhibit a clearly peaked structure as we incorporate more out-of-domain data for diversification. We demonstrated within a fixed budget, such trade-off indeed exists between specialization and enhanced instruction-following via training on a more diverse set of instructions. Figure 5b shows the trend when we diversify across an increasingly rich semantics. We notice the benefit of a more diverse $\mathcal{R}_{diver}$, which suggests that even when diversifying for specialists, one should be mindful to curate a dataset that spans over wider domains.

## 4 FINE-TUNING A SEPCIALIST INSTRUCTION-FOLLOWER: CASE OF CODE GENERATION

Building on our foundational insights from controlled string-rewriting tasks, we extend our analysis to training a real-world *specialist* instruction-follower.

Here, we investigate how cross-domain instruction diversity impacts the model's ability to handle real-world tasks that require a nuanced balance between instruction-following and domain-specific expertise by the example of code generation.

Code generation is primarily an instruction-following task, where models translate explicit prompts into executable code. Pre-trained code LLMs, having encountered common coding patterns during pre-training on large code corpora, are fine-tuned to utilize these structures effectively. Unlike reasoning tasks that demand creative problem-solving, code generation focuses on adhering to established patterns and passing test cases, emphasizing precise procedural execution over complex reasoning.

In this experiment, we aim to demonstrate that a diverse set of instructions can significantly enhance a model's adaptability to new coding instructions. This analysis provides crucial evidence for optimizing instruction-tuning datasets in real-world scenarios where both precision and generalization are paramount.

### 4.1 EXPERIMENTS

To rigorously evaluate the impact of cross-domain instruction diversity, we employ two widely-used code generation benchmarks: HumanEval (Chen et al., 2021a) and MBPP (Austin et al., 2021), alongside the augmented EvalPlus (Liu et al., 2023). These benchmarks present a diverse array of coding challenges that test the model's ability to interpret and execute novel instructions. As our training dataset, we use 20,000 instances sampled from the OSS-Instruct (Wei et al., 2023), a synthetic coding dataset, which has been sanitized to avoid data contamination. We also incorporate general-domain instruction data from Alpaca Taori et al. (2023), a well-known dataset that covers a

wide semantic domain. By gradually replacing code-specific instruction data with general-domain instructions, we assess how this cross-domain diversity influences the model's performance in code generation. We utilize two state-of-the-art pre-trained code language models as base models: DeepSeek-Coder-6.7B-Base (Guo et al., 2024) and CodeQwen-7B-Base (Bai et al., 2023).

## 4.2 STRIKING THE RIGHT BALANCE BETWEEN CODING AND DIVERSE DATA

**The Role of Semantic Diversity in Generalization**  Our results in Tables 2 and 1 highlight a crucial insight: while increasing the size of coding datasets may seem like the obvious solution for improving performance, this strategy is not always optimal. In fact, **diversifying instruction domains leads to greater improvements**. For example, adding data from general-purpose QA (Alpaca) and reasoning tasks (CoT) significantly outperforms incorporating an equivalent amount of coding data. Notably, the **Plum-colored** configuration in Table 1, which uses only 20,000 data points, surpasses the performance of models trained on 75,000 coding data points.

This pattern is consistent with our synthetic experiments in Section 3.2.2, where diverse instructions enabled the model to generalize better to out-of-distribution task instances for seen abstract instructions. Even in the basic string-replacement experiments (Section 2.3.4), we observed that introducing varied instructions—such as those in the Alpaca dataset—expanded the semantic range and boosted code generation performance, even when the quantity of each instruction is low compared to the main task (the long-tail distribution scenario in Section 2.3.3).

**The Power of Cross-Domain Diversification**  Figure 1a demonstrates how instruction-tuning datasets cluster within specific semantic sub-spaces. By incorporating datasets like Alpaca, which is designed for human language interaction, and the CoT-Collection Kim et al. (2023), which challenges the model with complex reasoning tasks, we extend the model's exposure to diverse semantic spaces. This further improves the model's generalization capabilities across domains.

Results from CodeQwen, presented in Table 2 support this conclusion. Models trained on a balanced mixture of coding, general QA, and reasoning data outperform those trained solely on a mix of coding data and Alpaca, reinforcing the importance of cross-domain diversification for handling complex and varied instructions.

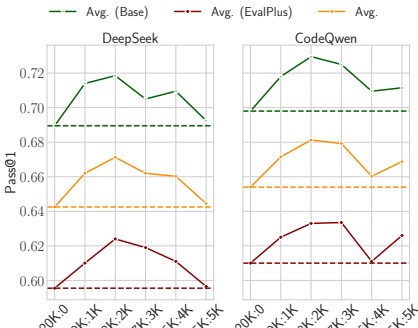

Figure 6: Sweet spots of Pass@1 with data mixture. Baseline is marked with dotted lines.

**Balancing Generalization and Specialization**  While diversifying instruction data offers substantial benefits, there are limits. As shown in Figure 6, incorporating general-domain instructions initially boosts the model's ability to follow natural language specifications, leading to improved Pass@1 scores for code generation. However, as more non-coding data is added, the model's capacity to handle the nuanced requirements of coding tasks diminishes.

This trade-off echoes our findings from the synthetic experiments in Section 3.2.2: achieving optimal performance requires balancing general instruction-following capabilities with specialized domain knowledge (Ling et al., 2024). Figure 6 illustrates the plateau and eventual decline in performance, underscoring the importance of calibrating the mixture of coding and non-coding data to achieve the best overall results.

The real-world experiment reinforces the insights from our synthetic experiments, emphasizing the importance of instruction diversity.

## 5 FINE-TUNING GENERALIST LLMS

In this section, we evaluate the impact of cross-domain instruction diversification on general reasoning tasks and investigate the optimal high-level strategy to improve the quality of a dataset.

| Base Budget (OSS) | + OSS | + Alpaca | + CoT | HE | HE+ | MBPP | MBPP+ | Avg (Base) | Avg (+) | Avg | Rel.Gain Intra -Budget | Rel.. Gain wrt. No Diver. |
|---|---|---|---|---|---|---|---|---|---|---|---|---|
| 19 | 1 | 0 | 0 | 65.9 | 59.8 | 73.7 | 62.2 | 69.8 | 61.0 | 65.4 | - | - |
| 19 | 0 | 1 | 0 | 67.7 | 61.6 | 75.9 | 63.4 | 71.8 | 62.5 | 67.2 | 2.8 | 2.8 |
| 18 | 0 | 2 | 0 | 69.5 | 63.4 | 76.4 | 63.2 | 73.0 | 63.3 | 68.1 | - | 4.1 |
| 18 | 0 | 1 | 1 | 68.9 | 63.4 | 77.4 | 64.2 | 73.2 | 63.8 | 68.5 | 0.6 | **4.7** |
| 16 | 0 | 4 | 0 | 65.2 | 58.5 | 76.7 | 63.7 | 71.0 | 61.1 | 66.0 | - | 0.9 |
| 16 | 0 | 2 | 2 | 68.3 | 61.6 | 77.2 | 63.7 | 72.8 | 62.7 | 67.7 | 2.6 | 3.5 |

Table 2: Pass@1 with CodeQwen-7B. We highlighted the accuracies surpassing baseline result with green and those falling below with red.

| Base Data Budget | +UI | +OO | +AL | Total | Overall Avg. | IF Avg. | Overall wo IF | Overall. wo Math | Rel. Gain + 20 Data Budget | Rel. Gain Diver. w./In Budget |
|---|---|---|---|---|---|---|---|---|---|---|
| | 10 | 0 | 0 | | 39.07 | 25.32 | 47.27 | 44.94 | - | - |
| **10** | 0 | 10 | 0 | **20** | **44.93** | **43.85** | 49.03 | 48.77 | - | 15.00 |
| | 0 | 0 | 10 | | 44.78 | 36.43 | **51.51** | **50.05** | - | 14.64 |
| | 20 | 0 | 0 | | 41.01 | 23.99 | 50.27 | 48.97 | 4.97 | - |
| **20** | 0 | 20 | 0 | **40** | 46.87 | 46.25 | 50.96 | 49.38 | 19.99 | 14.31 |
| | 0 | 0 | 20 | | 44.41 | 37.13 | 50.86 | 49.21 | 13.68 | 8.31 |
| | 10 | 10 | 0 | | **47.59** | **46.58** | **51.64** | **50.44** | 21.81 | 16.05 |
| | 10 | 0 | 10 | | 43.47 | 30.84 | 51.50 | 49.51 | 11.27 | 6.01 |
| | 20 | 0 | 0 | | 40.39 | 22.01 | 50.43 | 46.34 | -1.50 | - |
| **40** | 0 | 20 | 0 | **60** | 45.64 | **46.69** | 49.75 | **52.45** | 11.31 | 13.00 |
| | 0 | 0 | 20 | | 40.64 | 31.93 | 47.86 | 48.62 | -0.90 | 0.61 |
| | 10 | 10 | 0 | | **45.79** | 40.10 | **51.89** | 51.08 | 11.66 | 13.37 |
| | 10 | 0 | 10 | | 42.73 | 31.30 | 50.63 | 48.45 | 4.21 | 5.79 |
| | 20 | 0 | 0 | | 41.71 | 24.67 | 51.62 | 47.97 | 3.27 | - |
| **60** | 0 | 20 | 0 | **80** | 45.86 | **42.34** | 51.19 | 50.50 | 13.54 | 9.95 |
| | 0 | 0 | 20 | | 43.27 | 33.02 | 50.83 | 48.39 | 7.14 | 3.75 |
| | 10 | 10 | 0 | | **46.07** | 40.82 | **51.93** | 51.17 | 14.06 | 10.45 |
| | 10 | 0 | 10 | | 42.66 | 30.58 | 50.49 | 47.36 | 5.63 | 2.29 |

Table 3: The table shows the performance of generalist models trained with different data mixtures. **UI** refers to *UltraInteract*, **OO** refers to *OpenOrca*, and **AL** refers to *Alpaca*. The column labeled **Rel. Gain + 20 Data** indicates the relative performance gain of the model in the current row compared to the **UI**-only baseline that uses 20 fewer data points. For example, the performance of a model trained on **40** data will be compared to the baseline model trained on **20 UI** data, as indicated by the blue row above the current data quantity.**Rel. Gain Diver.** denotes the gain of diversification compared to the baseline with the same data budget.

## 5.1 EXPERIMENTAL SETUP

In this study, we evaluate the impact of cross-domain instruction diversification on large language models (LLMs) by comparing our approach with a baseline model trained exclusively on UltraInteract-SFT dataset (Yuan et al., 2024). UltraInteract-SFT is a collection of complex, multi-step reasoning problems emphasizing on math problem-solving, code generation, and logical reasoning, promoting robust reasoning and planning capabilities in LLMs.

While UltraInteract-SFT primarily focuses on math and coding problems and contains a rich collection of those problems, its scope is limited to these domains. OpenOrca (Lian et al., 2023) and Alpaca, though sparse and varied, introduce broader instruction-following tasks.

To assess the effectiveness of cross-domain instruction diversity, we constructed a training set that includes a mixture of UltraInteract-SFT, OpenOrca, and Alpaca datasets. While UltraInteract-SFT is rich in math, coding and complex QA problems, it remains limited to primarily these domains despite its challenging and diverse nature within them. OpenOrca and Alpaca, on the other hand, introduce instruction-following tasks across a broader range of domains, enriching the training data with varied instruction types. We gauged the model's overall capabilities using the same set of benchmarks consisting of coding (Austin et al., 2021; Chen et al., 2021a), math (Hendrycks et al., 2021b; Cobbe et al., 2021; Chen et al., 2023), knowledge (MMLU Hendrycks et al., 2021a), instruction following (Zhou et al., 2023b) and chain-of-thought reasoning (Suzgun et al., 2022) as (Yuan et al., 2024) and computed average performance.

To reflect on its precision in instruction following, we adopted IF-Eval (Zhou et al., 2023b) benchmark, comprising over 500 prompts for rigorous instruction-following tests. We followed a core-set selection approach when curating datasets of various budgets.

## 5.2 Data Diversity Matters More Than Quantity For Generalists

Our findings emphasize that expanding the model's exposure to varied domains leads to superior overall performance, underscoring the importance of data variety over sheer volume in enhancing model robustness and adaptability. Table 3 demonstrates the clear advantage of training models with diversified data across different domains, particularly when working with various data budgets. The key takeaway is that models exposed to a broad range of domains consistently achieve better performance than those trained solely on domain-specific reasoning data. This holds true even when tested on tasks that demand strong reasoning skills, which reinforces the results discussed in Section 3.2.1.

The results suggest that when aiming to improve a model's overall capability through dataset expansion, it's more effective to prioritize diverse datasets rather than simply increasing the volume of data from a specific domain. Our findings highlight that exposing models to varied domains enhances their overall performance, emphasizing the importance of diversity in training data for building robust and adaptable models, compared to focusing on dataset size alone.

## 6 Related works

**Datasets for instruction-tuning.** Many datasets for instruction-tuning have been proposed. The best quality is achieved for sets collated by human annotators Khashabi et al. (2020); Ye et al. (2021); Sanh et al. (2022b); Wang et al. (2022); Longpre et al. (2023); Conover et al. (2023); Köpf et al. (2023), but their size is constrained by the cost of annotation. Alternative methods, which use large language models to generate instruction sets, have been proposed Wang et al. (2023b); Honovich et al. (2022); Taori et al. (2023); Peng et al. (2023); Chiang et al. (2023); Xu et al. (2023a); Köksal et al. (2023); Kim et al. (2023). They provide larger instruction sets, at the cost of reduced annotation quality.

**Data curation for instruction-tuning.** It is widely recognized that the quality of instruction-tuning datasets has a massive impact on the performance of fine-tuned models. Previous works acknowledged the contributions of several key factors. Most research on the subject insist on the importance of the size and quality of the instruction sets (Chung et al., 2022; Iyer et al., 2022; Wang et al., 2023a). Liang et al. (Liang et al., 2024) point out the importance of consistent formats. Several recent works Zhou et al. (2023a); Cao et al. (2023) suggest that models fine-tuned on carefully selected examples can achieve high performance with small datasets. Various strategies for data curation have been proposed, focusing on instruction diversity, and the quality of answers (Zhou et al., 2023a; Cao et al., 2023; Xu et al., 2023b; Li et al., 2024; Liu et al., 2024b). Several authors discuss the benefit of mixing tasks from different categories Longpre et al. (2023); Iyer et al. (2022); Bukharin & Zhao (2024). Closest to our work, Dong et al. (Dong et al., 2024) discuss the impact of mixing general and domain-specific instructions, in order to achieve the best results with the smallest dataset.

## 7 Conclusion

In this work, we systematically studied instruction-following of language models via a suite of carefully designed controlled experiments. By introducing a symbolic string rewrite framework, we provide a focused model to isolate and study models' instruction-following abilities, offering a new perspective on this fundamental capability. Our experiments further demonstrate that instruction diversity, even within a fixed data budget, plays a critical role in improving model generalization. This finding underscores the value of diverse instruction semantics over large dataset size, in enhancing performance across both specialized and generalized LLM applications. Finally, we offer practical insights into dataset collation strategies, highlighting that proper diversification can significantly outperform dataset expansion.

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

## A  COMPLEMENT ON MARKOV ALGORITHMS

Markov algorithms Markov (1954) are ordered sets of rewrite rules, operating on sequences of symbols in a fixed alphabet $\mathcal{U}$. A sequence $S$ is processed by applying the first rewrite applicable to $S$, at the leftmost position if several exist: i.e. the rewrite rule $ss \to tr$ transforms the sequence $S = mississipi$ into $S' = mitrissipi$. The algorithm is then applied to $S'$, and the process is repeated until either no rules apply, and the algorithm is said to be *blocked*, or a special rule, called a *stop rule* is invoked, and the algorithm terminates and returns the final rewritten sequence.

Specifically, the algorithm uses and alphabet $\mathcal{A}$, which includes the alphabet $\mathcal{U}$ used buy the sequences to be processed (henceforth, small case latin letters), a set of additional symbols (henceforth, the small case greek letters $\{\alpha, \beta \dots\}$), and a special symbol $\cdot$ indicating a stop rule.

For instance, we could define the following algorithm, with $\mathcal{U} = \{a, b\}$, and $\mathcal{A} = \{a, b, \alpha, \beta, \cdot\}$, and the rules

$$
\begin{align}
\alpha x &\to x\alpha\beta x \tag{1}\\
\beta xy &\to y\beta x \tag{2}\\
\alpha\beta x &\to x\alpha \tag{3}\\
\alpha &\to \cdot \tag{4}\\
&\to \alpha \tag{5}
\end{align}
$$

where $x$ and $y$ stand for any letter $a$ or $b$. This will transform any sequence of $a$ and $b$ into a concatenation of the sequence and its reverse. Applied on $abb$, the algorithm will perform the following rewrites:

$$
\begin{align*}
abb &\to \alpha abb & \text{(by 5)}\\
\alpha abb &\to a\alpha\beta abb & \text{(by 1)}\\
a\alpha\beta abb &\to a\alpha b\beta ab & \text{(by 2)}\\
a\alpha b\beta ab &\to ab\alpha\beta b\beta ab & \text{(by 1)}\\
ab\alpha b\beta b\beta ab &\to ab\alpha\beta bb\beta a & \text{(by 2)}\\
ab\alpha\beta bb\beta a &\to ab\alpha b\beta b\beta a & \text{(by 2)}\\
ab\alpha b\beta b\beta a &\to abb\alpha\beta b\beta b\beta a & \text{(by 1)}\\
abb\alpha\beta b\beta b\beta a &\to abbb\alpha\beta b\beta a & \text{(by 3)}\\
abbb\alpha\beta b\beta a &\to abbbb\alpha\beta a & \text{(by 3)}\\
abbbb\alpha\beta a &\to abbbba\alpha & \text{(by 3)}\\
abbbba\alpha &\to abbbba & \text{(by 4)}
\end{align*}
$$

Since rule 4 is a stop rule, the algorithm terminates and returns $abbbba$.

Judicious introduction of additional (greek) letters allows one to compose Markov algorithms, effectively writing complex programs. Any effective process (i.e. finite computation) can be represented as a Markov algorithm (this is Markov's thesis).

## B  EXPERIMENTAL SET-UP

### B.1  MODEL AND TRAINING

In rewrite experiments, we train GPT-2 models Radford et al. (2019), a decoder-only transformer-based architecture, with 6 layers, 256 dimensions and 4 attention heads from scratch, on a generated instruction-tuning dataset using standard supervised fine-tuning approach. We use the AdamW optimizer, a learning rate of $10^{-3}$, and linear scheduling. All models are trained for 50 epochs. For the encrypted-rewriting task, we LoRA fine-tuned Llama-2 models with a learning rate of 1e-5, batch size 64, gradient accumulation step 1, and 8-bit quantization. The model takes about 2000 steps to

converge. For coding experiments, we trained the model with a learning rate of 1e-5, batch size 4, and gradient accumulation step 1, 8-bit quantization for 3 epochs with a maximum length of 768. The models are trained and inferenced on 1 Nvidia A40 GPU. We used greedy decoding for all experiments.

## B.2 DATA GENERATION

**Synthetic Experiment**   Except for the diversity of semantics experiment, the results we reported in the main paper are obtained from an input length of 50 and a pattern length of 20. To validate the generality of our findings, we conducted experiments on various input sizes {50, 100, 200} and, correspondingly, pattern lengths {20,40,50}.

In the diversity of semantics experiment, we used an input length of 500 and a pattern length of 60. We strictly restricted the sub-strings to look for and to replace them with both to be unseen during testing.

**Real World Data**   We downloaded the data from the official Huggingface Datasets Repos.

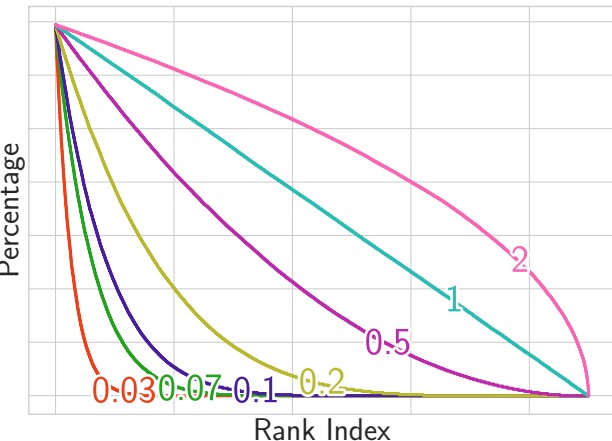

Figure 7: The sorted percentage of each instruction following power-law distribution with different shape parameters. The y-axis is the percentage of the rules in the training mixture. The x-axis is the ranked index (by proportion of examples) of instructions.

**Demonstration of dataset sizes for long-tail rule distribution experiments.**

## C   MORE DETAILS ON EVALUATION

For coding task, we evaluated the performance following the standard settings in EvalPlus Liu et al. (2023).

In Section 5, we evaluated a on variety of tasks. We used the evaluation suite (prompt, score computation script) provided by Yuan et.al (Yuan et al., 2024).

