# OpenReview forum: "$\textbf{Only-IF}$: Revealing the Decisive Effect of Instruction Diversity on Generalization"
_ICLR.cc/2025/Conference — ICLR 2025 Conference Withdrawn Submission_

### Official Review · Reviewer_x9at · 2024-11-02

**Soundness:** 2
**Presentation:** 3
**Contribution:** 2
**Rating:** 5
**Confidence:** 3

**Summary:**

This paper investigates the factors influencing the instruction-following capabilities of LLMs. To carry out the experiments, the authors design a series of controlled settings, including string replacements, rewriting with abstraction, and code generation. The results indicate that when the data budget is fixed, the diversity of instructions plays a vital role in enhancing model generalization. The authors focus on the impact of instruction diversity on performance. They design multiple scenarios, ranging from simple string replacements to complex code generation, to conduct their experiments. The analysis for the experiments is detailed.

**Strengths:**

1. The experimental setup is cleverly designed, making it easy to generate the data for experiments, and the tasks are well-suited for testing the instruction-following capabilities of LLMs.

2. A substantial number of controlled experiments are conducted to examine the instruction-following capabilities of LLMs, with a clear and detailed experimental setup.

**Weaknesses:**

1. The authors argue that using diverse instructions to train models can enhance performance. However, as illustrated in Table 1, the performance of the configuration with a Base Budget of 60, +OSS(14), +Alpaca(0.5), and +CoT(0.5) is lower than that of the configuration with a Base Budget of 60, +OSS(14), +Alpaca(1), and +CoT(0). Similarly, the performance of the configuration with a Base Budget of 60, +OSS(0), +Alpaca(7.5), and +CoT(7.5) is lower than that of the configuration with a Base Budget of 60, +OSS(0), +Alpaca(15), and +CoT(0). These findings do not align with the authors' argument.

2. The results of combining all three different domains (e.g., +UI(6), +OO(6), +AL(4)) are missing. Exploring this setting would provide interesting insights.

**Questions:**

The results of combining all three different domains (e.g., +UI(6), +OO(6), +AL(4)) are missing. Exploring this setting would provide interesting insights.

---

### Official Review · Reviewer_xBRf · 2024-11-02

**Soundness:** 2
**Presentation:** 2
**Contribution:** 2
**Rating:** 3
**Confidence:** 4

**Summary:**

The paper investigates the role of instruction diversity in enhancing large language models' (LLMs) generalization capabilities. The authors conduct controlled experiments inspired by the Turing-complete Markov algorithm, showing that generalization to unseen instructions is only achievable with training data diversified across various semantic domains. Their findings reveal that simply expanding data within a single domain does not guarantee robust generalization. Instead, cross-domain data diversification, even with limited data, significantly improves adaptability. The paper extends its analysis to real-world applications, highlighting that both specialist and generalist models benefit from instruction diversity. Specialist models, when trained with data beyond their core domain, show improved performance, while generalist models excel with a mixture of diverse instructions, enhancing their ability to follow instructions across tasks. This study emphasizes the importance of strategic data diversification over sheer dataset size for optimizing model performance and offers guidelines for effective dataset collation to support broad instruction-following capabilities in LLMs

**Strengths:**

+ The study challenges the assumption that merely expanding data volume improves performance, highlighting the superior benefits of semantic diversity in training data.
+ Practical guidance on dataset collation is offered, making this paper valuable for optimizing LLMs for a wide range of applications.
+ Overall, the work emphasizes the importance of instruction diversity, contributing actionable insights for building more robust, generalizable language models.

**Weaknesses:**

+ The setting is weird here: Why should we isolate the IF from the reasoning? On my understanding, these are two joint capabilites, and instruction following can be part of the reasoning. Moreover, IF is a general capability, which is supposed to include many other abilities such as coding, math, etc.
+ I have a strong concerns on the section 2.2, where they train a GPT-2 on some easy tasks for real LLMs, e.g., LLaMA-3.1-70B-Instruct. GPT-2 is a pre-LLM-era model and sft it does not reflect anything. I suggest the author using the LLaMA-70B models as the baseline and finetune these models to see if it can have the similar results/conclusions, which make more sense since the ultimate goal of this paper is to have a clear guideline for improving data quality/highlighting the critical role of diversity for future LLM sft/designs.
+ For the section 4.2, they conduct some experiments on coding, which yields very similar findings with the previous works [1, 2]. I don't think the contributions they claim is strong here. I am expecting a more interesting/impressive findings.
+ Overall speaking, the paper does not meet the bar of ICLR.



[1] Bukharin, A. and Zhao, T., 2023. Data diversity matters for robust instruction tuning. arXiv preprint arXiv:2311.14736.

[2] Zhang, X., Chen, Z.Z., Ye, X., Yang, X., Chen, L., Wang, W.Y. and Petzold, L.R., 2024. Unveiling the Impact of Coding Data Instruction Fine-Tuning on Large Language Models Reasoning. arXiv preprint arXiv:2405.20535.

**Questions:**

Please see weaknesses.

---

### Official Review · Reviewer_recE · 2024-11-02

**Soundness:** 2
**Presentation:** 2
**Contribution:** 2
**Rating:** 3
**Confidence:** 4

**Summary:**

This paper conducts a comprehensive empirical study to justify a core point of having a diverse set of training data enhances both specialist and generalist models. In particular, the authors design a toy string replacement experiment to decouple the model's core instruction-following capability with other capabilities as a controlled study. Further experiments are conducted to support the main point that diversified instruction tuning data benefits both generalist and specialist LLMs.

**Strengths:**

The main point that data diversification benefits LLM performance makes sense, as also pointed out by prior works.
The string-based toy experiments are interesting.

**Weaknesses:**

As an empirical study, the authors tries to make a point about the importance of data diversification. I have no doubt about the point, but would like to point out the following:

1) The viewpoint is not new - if we check foundation models's tech reports, we can already observe that their instruction tuning data is very diverse. Then, what's the additional insight?
2) Given that, what I expect is not to prove the correctness of the claim, but a more systematic and clear recipe to collect and curate such diverse data. For example, for generalist models, how much data points we need in the instruction tuning set, what will be the recommended (if not the optimal) distribution of the instruction tuning data, including domain, length, use case, persona, etc. Unfortunately, I still have no idea after reading the paper.
3) Although the string-based toy experiments are interesting, can we really make any conclusion out of them? For example, in 2.2, the author train GPT-2 models, then how to make sure that any observation scales with larger model sizes and newer architecture family?
4) Even for more realistic settings, I would expect a more comprehensive experiments with larger LLMs beyond 7B - a natural questions is, for larger models trained on (possibly) larger and more diverse corpus, does the diversity of instruction tuning data matter as much?

Finally, I would like to justify my points a bit: as an empirical study paper with a strong claim, the reviewer naturally wants to see clear and strong evidence to support the claim. I am not asking for experimenting with all types of LLMs, which is apparently not possible. But at least, more model sizes and settings should be considered before making such a general conclusion. However, the story will be very different if there were any theoretical backups.

**Questions:**

In addition to questions in the weaknesses part, I have the following question:
1) I may miss the information - what is the base LLM in Section 5 Fine-tuning Generalist LLMs?

---

### Official Review · Reviewer_VRx2 · 2024-11-04

**Soundness:** 3
**Presentation:** 3
**Contribution:** 2
**Rating:** 5
**Confidence:** 4

**Summary:**

The paper explores how instruction generalization in large models emerges when the training data is sufficiently diverse across different semantic spaces.It conducts experiments with specialist and generalist models: 1)Training specialists with data that extends beyond their core domain leads to significant performance improvements. 2)Generalists benefit from a well-balanced mix of diverse data.
The study emphasizes the critical role of data diversity in enhancing overall data quality.

**Strengths:**

- The focus on the trade-offs and dynamics between specialists and generalists is both relevant and compelling.
- The experimental design is well thought out and effectively supports the conclusions drawn.

**Weaknesses:**

- The link between the research gap and the experimental findings is unclear. The authors discuss the “decomposition of reasoning and knowledge retrieval in LLMs” in Line 51, but then shift to emphasizing data diversity for specialists and generalists in Lines 80-88. How do these concepts directly connect? The introduction would benefit from reorganization to clarify this relationship.
- The paper’s structure is difficult to follow. Providing a brief overview of the experimental setup at the end of the introduction could improve readability and coherence.
- The layout of content and figures is confusing. For instance, Figure 1 appears on Page 1 but isn’t referenced until Page 8. It would help to either highlight the main insights from Figure 1 in the introduction or improve figure placement.
- While the results on specialists and generalists are insightful, more quantitative analysis would strengthen the paper. For example, how much general data is needed to enhance specialist models effectively for a given task?
- In Section 5, the authors claim that greater domain diversity enhances generalists. However, it’s unclear how they controlled for the impact of domain scaling versus data size. How can these two factors be disentangled?

**Questions:**

- Some relevant references on training specialist and generalist models seem to be missing [1-2]. These studies emphasize the importance of mixing instructions from both general and specific domains and could strengthen your argument.
-  In Figure 6, the results for CodeQwen lack comprehensive coverage regarding the impact of scaling mixtures of specific and general data. What additional findings would emerge if more diverse data were incorporated?
- Minor typos: Line 50: Replace with the correct citation format: (Dong et al., 2024; Zhang et al., 2024b).

[1] Yuan, Lifan, et al. “Advancing LLM Reasoning Generalists with Preference Trees.” arXiv preprint arXiv:2404.02078 (2024).

[2] Zhang, Kaiyan, et al. “Ultramedical: Building Specialized Generalists in Biomedicine.” arXiv preprint arXiv:2406.03949 (2024).

---

### Note · Authors · 2024-12-04

I have read and agree with the venue's withdrawal policy on behalf of myself and my co-authors.